# Acrylamide–Hemoglobin Adduct Levels in a Japanese Population and Comparison with Acrylamide Exposure Assessed by the Duplicated Method or a Food Frequency Questionnaire

**DOI:** 10.3390/nu12123863

**Published:** 2020-12-17

**Authors:** Junpei Yamamoto, Junko Ishihara, Yasuto Matsui, Tomonari Matsuda, Ayaka Kotemori, Yazhi Zheng, Daisuke Nakajima, Miho Terui, Akiko Shinohara, Shuichi Adachi, Junko Kawahara, Tomotaka Sobue

**Affiliations:** 1School of Life and Environmental Science, Azabu University, 1-17-71 Fuchinobe, Chuo-ku, Sagamihara, Kanagawa 252-5201, Japan; j-yamamoto@azabu-u.ac.jp (J.Y.); kotemori@azabu-u.ac.jp (A.K.); 2Graduate School of Engineering, Kyoto University, Yoshida Honmachi, Sakyo-ku, Kyoto 606-8501, Japan; ymatsui@risk.env.kyoto-u.ac.jp (Y.M.); matsuda.tomonari.8z@kyoto-u.ac.jp (T.M.); 3Division of Epidemiology, Center for Public Health Sciences, National Cancer Center, 5-1-1 Tsukiji, Chuo-ku, Tokyo 104-0045, Japan; 4Center for Health and Environmental Risk Research, National Institute for Environmental Studies, 16-2 Onogawa, Tsukuba, Ibaraki 305-8506, Japan; yazhizheng@yahoo.co.jp (Y.Z.); dnakaji@nies.go.jp (D.N.); junkawa0227@yahoo.co.jp (J.K.); 5Department of Public Health, Faculty of Nutritional Science, Sagami Women’s University, 2-1-1 Bunkyo, Minami-ku, Sagamihara, Kanagawa 252-0383, Japan; goldear118@gmail.com (M.T.); akikon48128@gmail.com (A.S.); s-adachi@star.sagami-wu.ac.jp (S.A.); 6Department of Environmental Medicine and Population Sciences, Graduate School of Medicine, Osaka University, 2-2 Yamadaoka, Suita, Osaka 565-0871, Japan; tsobue@envi.med.osaka-u.ac.jp

**Keywords:** acrylamide, hemoglobin adducts of acrylamide, duplicate method, food frequency questionnaire

## Abstract

The levels of hemoglobin adducts of acrylamide (AA–Hb), a biomarker of acrylamide exposure, have not been reported for Japanese subjects. Herein, we determined the AA–Hb levels in a Japanese population and compared them with the estimated dietary intake from the duplicate diet method (DM) and a food frequency questionnaire (FFQ). One-day DM samples, FFQ, and blood samples were collected from 89 participants and analyzed for acrylamide. AA–Hb was analyzed using liquid chromatography tandem mass spectrometry and the N-alkyl Edman method. Participants were divided into tertiles of estimated acrylamide intake and geometric means (GMs) of AA–Hb adjusted for sex and smoking status. A stratified analysis according to smoking status was also performed. The average AA–Hb levels for all participants, never, past, and current smokers were 46, 38, 65, and 86 pmol/g Hb, respectively. GMs of AA–Hb levels in all participants were significantly associated with tertiles of estimated acrylamide intake from DM (*p* for trend = 0.02) and FFQ (*p* for trend = 0.04), although no association with smokers was observed. AA–Hb levels reflected smoking status, which were similar to values reported in Western populations, and they were associated with estimated dietary intake of acrylamide when adjusted for sex and smoking status.

## 1. Introduction

Acrylamide is a water-soluble chemical compound mainly produced in starchy foods during processing and cooking via a chemical reaction between asparagine and reducing sugars at high temperatures [1,2,3]. Tobacco smoke is also a major source of acrylamide exposure [4,5]. The International Agency for Research on Cancer classifies acrylamide as a group 2A carcinogen that is “probably carcinogenic to humans” [6].

In the EU, research on acrylamide in food and its risk assessment is being carried out vigorously and has been leading in the world. The European Food Safety Authority (EFSA) released its scientific opinion on acrylamide in food [7]. The EFSA conducted the monitoring of acrylamide levels in food from 2007 to 2010 and reported the analysis of 13,162 samples in 25 countries covering 10 different food categories [8]. Based on the monitoring, the Europe Commission set indicative values for acrylamide in various foodstuffs [9]; for instance, the value in potato crisps was 1000 µg/kg. The EU Commission also published an acrylamide database containing data on the amount of acrylamide in food ingredients [10]. In Japan, the Food Safety Commission plays a central role in monitoring and risk assessment of acrylamide in food [11]. In addition, a number of studies aimed at reducing acrylamide content in food were reported [12,13]. Consequent to the publication of these studies and owing to the efforts of food manufacturing companies, the median concentration of acrylamide in potato crisps produced in Japan decreased from 940 in 2004 to 550 μg/kg in 2015 [11]. In addition, a database for calculating the acrylamide intake in Japanese individuals from dietary assessment was developed by Kotemori et al. in 2018 by consolidating several reports monitoring acrylamide levels in Japanese foods [14].

Acrylamide is a reactive molecule that forms adducts with the N-terminal valine in hemoglobin (called the “AA–Hb adduct”), and the concentration of AA–Hb is strongly correlated with the levels of acrylamide exposure [15,16,17]. For example, tobacco smoking, occupational exposure, and tobacco smoke present in the environment can influence the levels of AA–Hb [18]. Smokers have three- to four-times higher levels of AA–Hb than non-smokers [19]. Therefore, AA–Hb is considered a relevant biomarker of internal exposure to acrylamide and represents exposure over the life span of erythrocytes in the previous 4 months [20,21]. Additionally, AA–Hb has been used as a biomarker in multiple epidemiological studies. Recently, AA–Hb levels were measured in about 20 epidemiological studies. In the largest study (*n* = 7166), AA–Hb levels were reported to range from 3 to 910 pmol/g hemoglobin, the geometric mean (GM) of AA-Hb levels was 50.0 pmol/g hemoglobin in non-smokers (*n* = 5686) and 113 pmol/g hemoglobin in smokers (*n* = 1282) in the general US population [18].

In Japan, Kotemori et al. and Liu et al. revealed that dietary acrylamide intake was not associated with breast, endometrial, ovarian, esophageal, gastric, or colorectal cancer in a large-scale population-based prospective cohort study [22,23,24]. These reports showed that dietary acrylamide exposure might not increase the risk of cancer in Japanese individuals. However, levels of acrylamide intake were estimated only using food frequency questionnaire (FFQ), and thus, these associations may be attenuated by measurement errors of FFQ.

AA–Hb levels have also been used for validation of the estimated acrylamide intake assessed by an FFQ. Wilson et al. reported a moderate correlation between acrylamide intake estimated by FFQ and AA–Hb in non-smokers (correlation coefficient (CC) = 0.25) [25], and similar results were observed in another study.

Four nested case-control studies were performed to evaluate the association between acrylamide exposure and the risk of cancer using AA–Hb [26,27,28,29]. Olesen et al. reported that AA–Hb levels were positively associated with estrogen receptor-positive breast cancer after adjusting for the smoking behavior (incidence rate ratio (IRR) = 2.7; 95% confidence intervals (CI), 1.1–6.6) [28]. However, other studies have shown no statistically positive associations [26,27,29]. No epidemiological studies have been conducted to evaluate the association between AA–Hb levels and cancer in a Japanese population. Before investigating these associations, a correlation between AA–Hb levels and estimated acrylamide intake from dietary surveys must first be investigated.

Accordingly, in this study, we aimed to measure AA–Hb levels in Japanese individuals using international standard methods. We also evaluated the association between AA–Hb levels and estimated acrylamide intake obtained from the diet research method.

## 2. Materials and Methods

### 2.1. Study Participants

A total of 119 volunteers participated in this study between 2015 and 2016 [30]. Participants were aged 20 years or older and were living in the Tokyo, Ibaraki, or Kanagawa prefecture, Japan. We recruited participants through website messages and flyer distribution so that gender age groups (20–29, 30–39, 40–49, 50–59, and ≥60 years-old) would be evenly distributed. The details of the study were explained to the participants, and informed consent and individual information from all volunteers were obtained.

### 2.2. Data Collection

Food samples for the duplicate method (DM) were collected as described elsewhere [30]. Briefly, participants saved the duplicates of all foods and drinks (including mixed dishes) they consumed during the collection day (any day without specifying weekdays or holidays). Each food and drink sample was separately placed in plastic bags or containers and stored in a refrigerator at the homes of the participants. After collection, samples were stored at −30 °C until analysis.

After collecting DM samples, we asked participants to answer an FFQ, which was designed to assess habitual dietary intake for the previous year. The FFQ was originally developed and validated for use in the Japan Public Health Center-based Prospective Study [31,32,33]. The FFQ consisted of 138 food and beverage items and nine frequency categories, ranging from “almost never” to “7 or more times per day” (or “9 glasses per day” for beverages) and contained questions regarding the usual consumption of foods during the previous year. Standard portion sizes were specified for each food item as follows: small (50% smaller), medium (same as the standard), and large (50% larger).

Blood samples (18 mL, 5 h-fasting) were collected in 2016 from participants. The time period between DM collection and blood sampling varied from 3 days to 3 months (or to 5 months for 11 participants). Prior to blood collection, the staff checked the health condition and medications of the participants through an interview and measured their height and weight. The samples were centrifuged at 3500–4000 rpm for 10 min and divided into erythrocytes, plasma, buffy coat, and serum. The samples were then stored at −80 °C until analysis.

### 2.3. Analysis of AA–Hb in Blood

Analysis of AA–Hb in erythrocytes was conducted using the N-alkyl Edman method, which is used to measure AA–Hb in many epidemiological studies [25,34,35]. Briefly, the purified protein fraction including hemoglobin in erythrocytes was obtained by ethanol precipitation following centrifugation with acetone. For Edman degradation, fluorescein isothiocyanate (FITC) was added to the protein fraction and incubated for 18 h. FITC reacts with the N-terminal amino acid (Val) in hemoglobin, releasing an FITC-labeled acrylamide–hemoglobin (Val) adduct (AA–Val–FTH). After incubation, the samples were purified by solid-phase extraction and analyzed using liquid chromatography tandem mass spectrometry (LC-MS/MS). Concentrations of AA–Val–FTH were corrected according to the level of acrylamide adducts with the N-terminal amino acid in hemoglobin and eight amino acid residues (AA–VHLTPEEK) that can correct differences in Edman degradation rates between samples. The lower limit of quantitation of AA–Val–FTH was 1.25 nM, and the percent coefficient of variation was 8.2. In addition, AA–Hb levels were adjusted by hemoglobin concentrations of erythrocytes that were measured using the QuantiChrom Hemoglobin Assay Kit (Bioassay Systems, Hayward, CA, USA) according to the manufacturer’s instructions.

### 2.4. Assessment of Acrylamide Exposure using DM and FFQ

DM is an analytical method for determining exposure from foods that provided information for 1-day exposure in this study. The quantification of acrylamide in food samples from the DM was performed by Kawahara et al. [30]. Briefly, 5 g of homogenized food sample was shaken with hexane and water for 60 min, and the aqueous layer was collected. The aqueous layer was cleaned up by solid-phase extraction with an Oasis hydrophilic–lipophilic balance (HLB) cartridge and an Oasis mixed–mode, cation–exchange (MCX) cartridge (Waters, Milford, MA, USA). The purified samples were analyzed with LC-MS/MS using an InertSustain AQ-C18 HP column (5 µm, 150 × 2.1 mm; GL Sciences Inc., Tokyo, Japan). The limit of quantification of acrylamide in the diet was 1.7 ng/g.

The FFQ was used to assess long-term exposure and indicated the habitual dietary intake for the previous year. The estimation of acrylamide intake using our FFQ was previously described [14]. Briefly, acrylamide intake was calculated using each food intake data from FFQ and a developed acrylamide content database. This database was developed by Kotemori et al. by integrating reports published on the amount of acrylamide present in food in Japan, published up to July 2016 [14]. Out of 147 food items in FFQ, 28 were designated as acrylamide-containing foods. In addition, the cooking method was considered for seven vegetables (onions, bean sprouts, sweet peppers, squash, cabbage, snap beans, and broccoli), potatoes, rice, bread, and fried batter. The proportions of each cooking method were extrapolated from the proportions calculated from the weighed record during the DM. In addition, the eating frequency of fried foods with batter was included as a question in our FFQ. Acrylamide intake from each food was calculated by multiplying the concentration of acrylamide for each food with the eating frequency and portion size. The total daily acrylamide intake from the FFQ was calculated by summing the acrylamide intake for each food. The de-attenuated correlation coefficients of energy-adjusted acrylamide intake estimated from our FFQ among men and women were 0.54 and 0.48, respectively, in Cohort I and 0.40 and 0.37, respectively, in Cohort II of the Japan Public Health Center validation study [14].

### 2.5. Statistical Analysis

FFQ and blood samples were collected from 106 of 119 (89.1%) and 102 of 119 (85.7%) participants, respectively. We excluded participants who did not submit FFQ and/or those who could not collect blood; a total of 89 participants were included in our study.

The means, standard deviations, and medians of AA–Hb and acrylamide intake from the DM samples and FFQ were calculated. Participants were divided into tertiles of acrylamide intake (T1–T3) according to acrylamide intake calculated from DM samples and FFQ, and GMs and 95% confidence intervals (CIs) of AA–Hb were calculated by back-transforming the arithmetic means of the log-transformed values adjusted for sex and smoking status. We further performed stratified analyses according to cigarette smoking status (current–past smoker or never smoker). Trend associations were assessed by assigning ordinal numbers 0–2 to three categories of acrylamide intake calculated from DM samples and FFQ.

### 2.6. Ethics Statements

This study was approved by the Institutional Review Board (IRB) of the National Institute for Environmental Studies (no. 1-2015-006) on 13 November 2015 and the IRB of Sagami Women’s University (no. 1404) on 29 May 2014. Additional IRB approval was provided by Osaka University (no. 15058) on 13 August 2015, by Sagami Women’s University (no. 1543) on 21 September 2015; by Azabu University (no. 112) on 4 August 2017; and by Kyoto University (no. R1211) on 25 October 2017. Written informed consent was obtained from the participants at the study orientation.

## 3. Results

### 3.1. Basic Characteristics of the Study Participants

The characteristics of the participants are presented in Table 1. The participants consisted of 89 healthy adults aged 20–78 years living near a metropolitan area in Japan. The age groups in the participants were almost equally distributed, and 60% of participants were women. Among the participants, 9% were current smokers, 11.2% were past smokers, and the remaining participants were never smokers. Body mass indexes were similar to the reference value for the Japanese population [36].

### 3.2. AA–Hb Levels in Erythrocytes and Estimated Acrylamide Exposure Levels from DM Samples and FFQ

Table 2 shows the average values for AA–Hb levels and acrylamide intake calculated from the DM and FFQ for all participants or stratified by smoking status (never smokers, past smokers, or current smokers). The average levels of AA–Hb for all participants, never smokers, past smokers, and current smokers were 45.7, 38.4, 65.3, and 85.5 pmol/g Hb, respectively. The average values for acrylamide intake calculated from the DM and FFQ were 225.2 and 130.6 ng/kg body weight (BW)/day, respectively, for all participants; 243.5 and 135.9 ng/kg BW/day, respectively, for never smokers; 157.3 and 102.7 ng/kg BW/day, respectively, for past smokers; and 147.6 and 120.2 ng/kg BW/day, respectively, for current smokers. AA–Hb levels were higher in past and current smokers, whereas the estimated acrylamide intake from DM and FFQ was higher in never smokers.

### 3.3. GM of AA–Hb Levels by Tertiles of Estimated Acrylamide Intake from DM and FFQ

Table 3 demonstrates the GMs of AA–Hb levels by tertiles of estimated acrylamide intake. After adjustment for sex and smoking status, the GMs of AA–Hb levels were significantly associated with tertiles of estimated acrylamide intake from DM (*p* for trend = 0.02) and FFQ (*p* for trend = 0.04). In the stratified analysis according to smoking status (never smokers, past smokers, or current smokers), AA–Hb levels were moderately associated with the estimated acrylamide intake in never smokers, although no association was observed in past or current smokers.

## 4. Discussion

In this study, we investigated the levels of AA–Hb as a biomarker of acrylamide exposure in 89 Japanese men and women using the N-alkyl Edman method with LC-MS/MS, the analytical methods used in previous epidemiological studies. Previous epidemiological studies in Western populations reported that AA–Hb levels ranged from 19 to 51 pmol/g Hb in non-smokers and from 80 to 194 pmol/g Hb in smokers [5,18,19,28,34,37,38,39,40,41,42,43]. In our study, AA–Hb levels were 45.7 pmol/g Hb in never smokers and 85.6 pmol/g Hb in current smokers (Table 2). Therefore, we found that AA–Hb levels among Japanese individuals were similar to those reported in Western populations and were similarly affected by the smoking status. To the best of our knowledge, this is the first study reporting the biomarker levels of acrylamide exposure in an Asian population.

The acrylamide intake estimated from the FFQ (131 ng/kg BW/day) was similar to the mean values (140 ng/kg BW/day) reported in the Japan Public Center-based Prospective Cohort Study (JPHC Study), which is one of the largest existing cohorts in Japan started in the 1990s [22,23]. Dietary acrylamide exposure levels estimated using FFQ were reported as 300 ng/kg BW/day in the Netherlands [44], 400 ng/kg BW/day in Norway [45], and 380 ng/kg BW/day in Sweden [46]. The estimated dietary intake of acrylamide in the Japanese population was lower than that in Western populations, as previously reported [14,47]. Similarly, acrylamide intake determined from the DM (225 ng/kg BW/day) in this study was lower than that in the study measuring intake in the Netherlands (450 ng/kg BW/day) [48]. Since AA–Hb levels are comparable to absolute exposure levels, Japanese subjects may have been exposed to more acrylamide from non-dietary factors than Western subjects. Acrylamide exposure reflects a combination of many factors, including intake from the diet, smoking, second-hand smoking, and drinking water [49]. Since we did not collect data for second-hand smoking, exposure from it was reflected in AA–Hb levels but not in the FFQ. In Japan, the Ministry of Health, Labor and Welfare has set the acrylamide concentration to less than 0.5 μg/L as the target value in tap water [50], although few reports have described the amount of acrylamide in tap water in Japan. These factors may have caused differences in estimation from the FFQ and measurement of AA-Hb levels compared with Western populations.

In addition, the estimated acrylamide intake from the FFQ may have been underestimated. We estimated acrylamide intake from the FFQ, using a database of acrylamide-containing foods [14] and considered the influence of home cooking methods for specific foods obtained from weighed records with the DM. However, the list of acrylamide-containing foods only covered approximately 19% of the food items in the FFQ. The DM is capable of overcoming these underestimations because, unlike a database, there are no missing values. Nonetheless, it is not appropriate to identify absolute acrylamide dietary exposure level using FFQ, and it is difficult to directly compare FFQ and DM because the FFQ reflects long-term exposure, and our DM samples reflect a single day of exposure. Therefore, we could not conclusively determine whether dietary intake levels were underestimated.

Ranking of acrylamide exposure by dietary acrylamide intake may be possible. GMs of AA–Hb levels were significantly associated with tertiles of estimated acrylamide intake from DM (*p* for trend = 0.02) and FFQ (*p* for trend = 0.04) after adjusting for the sex and smoking status (Table 3). In the stratified analysis according to the smoking status (never smoker, past smoker, or current smoker), GMs of AA–Hb levels were moderately associated with tertiles of DM and FFQ in never smokers, whereas no association with past and current smokers was observed owing to the small sample size. We also considered the association in smokers (by merging the past and current smoker) but did not find any association (data not shown). These findings suggest that acrylamide intake levels estimated by dietary assessment could be used to categorize acrylamide exposure by ranking after adjusting for the sex and smoking status. However, this may lead to misclassification within smokers and further studies including more smokers are required.

The potential cancer risk of acrylamide to the Japanese population could be higher than was previously expected. Since the dietary acrylamide exposure levels of Japanese individuals are lower than those in the Western population [14,47], the estimated margin of exposure (MOE) of dietary acrylamide for Harderian gland tumors in Japanese individuals (about 1000) is higher than that of EFSA (50–425) [7,11]. However, our study revealed that the AA-Hb level, which is a biomarker of acrylamide exposure, among Japanese individuals was similar to that in the Western populations. Therefore, an epidemiological approach using AA–Hb such as nested case-control studies is required to examine the cancer risk of acrylamide in the Japanese population.

There are some limitations to note. First, there were relatively few current smokers among the study participants. As a result of the heavy burden of DM, participants in our study had to be recruited on a volunteer basis, which may have lowered the ratio of the smokers. Further studies are needed to measure AA–Hb levels in the more general population. Second, AA–Hb levels were measured only once. AA–Hb levels represent exposure to acrylamide in the previous 4 months [20,21]. In contrast, the FFQ represents habitual dietary intake for the previous year. Therefore, by performing the AA–Hb measurement at multiple time points (e.g., every 4 months), comparisons with estimated values from the FFQ might be more appropriate. Third, the DM was conducted for only 1 day. DM may not have the ability to capture long-term acrylamide exposure. However, the DM could not be applied over multiple days in this study because of the considerable expenses involved and the burden on participants. To estimate habitual acrylamide exposure by DM, further studies are required to verify the day-to-day variation in acrylamide intake from food in the Japanese population.

## 5. Conclusions

We reported the levels of AA–Hb in an epidemiological study of Japanese individuals with a relatively low intake of acrylamide for the first time. AA–Hb levels reflected the smoking status and were similar to the values reported for the Western populations. The levels of AA–Hb were associated with the estimated acrylamide intake when adjusted for sex and smoking status, although no association with smokers was observed.

## Figures and Tables

**Table 1 nutrients-12-03863-t001:** Baseline characteristic of study participants.

Characteristics	Value
Participants, *n* Age, years (mean, SD)	89 42.3 (14.4)
20–29, *n*	22
30–39, *n*	15
40–49, *n*	22
50–59, *n*	18
60≥, *n*	12
Sex	
Men (%)	40.4
Women (%)	59.6
Smoking	
Current Smoker (%)	9.0
Past Smoker (%)	11.2
Never Smoker (%)	79.8
BMI, kg/m^2^ (mean, SD)	22.1 (3.6)
20<, *n*	24
20–25, *n*	48
25>, *n*	17

BMI, body mass index; SD, standard deviation.

**Table 2 nutrients-12-03863-t002:** Comparison of acrylamide exposure level from AA-Hb, DM, and FFQ.

Parameters	All (*n* = 89)	Never Smoker (*n* = 71)	Past Smokers (*n* = 10)	Current Smokers (*n* = 8)
AA-Hb (pmol/g Hb)	
Mean (SD)	45.7 (35.4)	38.4 (30.7)	65.3 (31.2)	85.5 (46.6)
Median	35.3	31.5	67.6	80.4
(5th %-ile, 95th %-ile)	(15.7, 104.3)	(15.7, 74.5)	(17.3, 118.9)	(13.8, 150.2)
DM (ng/kg bw/day)	
Mean (SD)	225.2 (267.8)	243.5 (288.0)	157.3 (140.5)	147.6 (173.4)
T1	50.9	59.7	37.7	22.0
T2	139.7	148.7	123.8	62.3
T3	479.2	514.5	321.7	316.7
FFQ (ng/kg bw/day)	
Mean (SD)	130.6 (79.4)	135.9 (81.2)	102.7 (60.4)	120.2 (84.4)
T1	60.8	66.0	53.5	40.4
T2	112.3	116.6	77.3	89.0
T3	218.8	225.0	185.8	204.5

AA-Hb, acrylamide hemoglobin adduct; SD, standard deviation; %-ile, percentile; DM, duplicate method; FFQ, food frequency questionnaire; T1–T3, tertiles of acrylamide intake estimated from DM or FFQ.

**Table 3 nutrients-12-03863-t003:** GMs and 95% CI of acrylamide hemoglobin adduct level according to tertile of acrylamide intake estimated from DM and FFQ.

Variables	All (*n* = 89)			Never Smoker (*n* = 71)			Past Smokers (*n* = 10)			Current Smoker (*n* = 8)		
	*n*	GM (95% CI) ^a^		*n*	GM (95% CI) ^a^		*n*	GM (95% CI) ^a^		*n*	GM (95% CI) ^a^	
**Tertile of each acrylamide intakes**												
**DM**												
T1	29	46.0 ^b^	(36.9–57.3)	20	29.2	(23.0–37.1)	5	48.6	(22.1–106.9)	4	63.8	(17.1–238.2)
T2	30	48.3 ^b^	(38.0–61.4)	25	29.3	(23.8–36.1)	3	61.1	(23.9–156.2)	2	62.5	(11.4–342.4)
T3	30	64.5 ^b^	(50.5–82.5)	26	38.0	(31.1–46.3)	2	86.4	(28.0–266.2)	2	97.2	(13.0–726.8)
*p* for trend		0.02			0.08			0.34			0.68	
**FFQ**												
T1	29	44.4 ^b^	(35.4–55.8)	21	27.8	(22.2–34.7)	4	52.9	(27.3–102.4)	4	69.2	(18.9–254.0)
T2	30	52.5 ^b^	(41.2–66.9)	25	33.6	(27.3–41.2)	3	39.0	(15.6–97.8)	2	94.0	(17.6–503.4)
T3	30	59.9 ^b^	(47.0–76.3)	25	36.2	(29.3–44.7)	3	97.0	(44.4–211.9)	2	54.9	(7.5–399.9)
*p* for trend		0.04			0.09			0.20			0.82	

GMs, geometric means; CI, confidence interval; DM, duplicated method; FFQ, food frequency questionnaire; T1–T3, tertiles of acrylamide intake estimated from DM or FFQ. ^a^ GMs were calculated by back transforming the arithmetic means of the log-transformed values adjusted for sex. ^b^ Adjusted for sex and smoking status.

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
