# Peer review of "Acrylamide–Hemoglobin Adduct Levels in a Japanese Population and Comparison with Acrylamide Exposure Assessed by the Duplicated Method or a Food Frequency Questionnaire"

_nutrients, 2020, doi:10.3390/nu12123863_

Round 1

Reviewer 1 Report

The article is very well written. The study is well designed and methods and results clearly described. The discussion is appropriate. The study added acrylamide exposure data in the Japanese people to the collection of global exposure. Further it illustrated a closer correlation of the biomarker with the DM based exposure than the FFQ based exposure, and confirm the contribution of smoking status to total exposure levels. I find little error in the article.

One weakness is the small number of the current and previous smokers that compromised the finding. Has the author considered to merge the two smoker groups in one group for the trend test? It is also necessary to discuss the potential public health risk with the level of exposure. Finally, the conclusion is not a good summary of the findings of the study, and should be rewritten.

Other minor comments

L 66 and L 71: add the correlation coefficient of each study please.

L 84: were health status (metabolic disorder for example) considered in the recruiting criteria?

L91: why are days not specified for weekday and weekend? It is important to know whether it is a routine or weekend meal.

L132: give the reference of this previous study

L134: what kind of database is it, give more detail or a reference to that.

Reviewer 2 Report

The manuscript entitled “Acrylamide–Hemoglobin Adduct Levels in a Japanese Population and Comparison with Acrylamide Exposure Assessed by the Duplicated Method or a Food Frequency Questionnaire” is a quality research about a substance that is a public concern. This manuscript have enough quality to be published in this journal. However, there are some issues that authors should include in this manuscript to improve it before its publication.

In the introduction section, I miss some explanations. In my opinion, authors should include European legislation and levels as EFSA Scientific Reports. EU food control and safety management is a reference to the world, its values and the monitoring process should be included and compared with the exposed in the manuscript. It is necessary to include some data and references to the following documents:

  • EFSA (2015) Scientific Opinion on acrylamide in food. EFSA Journal 13(6): 4104.
  • European Commission. Acrylamide Database. Available online: https://ec.europa.eu/food/safety/chemical_safety/contaminants/catalogue/acrylamide_db_en
  • COMMISSION RECOMMENDATION of  2  June  2010 on  the  monitoring  of  acrylamide  levels  in  food. Official Journal of the European Union. L 137/4.
  • RECOMMENDATIONS COMMISSION. COMMISSION RECOMMENDATION of 3 May 2007on the  monitoring of acrylamide levels in food(notified under document number C(2007) 1873) (Text with EEA relevance)(2007/331/EC). Official Journal of the European Union. L 123/33.

According to the method of analysis, authors should include this references to Japanese authors previously published and of interest to this manuscript:

  • Yoshida M. et al. (2005) Acrylamide in Japanese Processed Foods and Factors Affecting Acrylamide Level in Potato Chips and Tea. In: Friedman M., Mottram D. (eds) Chemistry and Safety of Acrylamide in Food. Advances in Experimental Medicine and Biology, vol 561. Springer, Boston, MA. https://doi.org/10.1007/0-387-24980-X_31
  • Yasui, A. New food control system in Japan and food analysis at NFRI. Accred Qual Assur 9, 568–570 (2004). https://doi.org/10.1007/s00769-004-0844-8

Reviewer 3 Report

The publication lacks a detailed description of the acrylamide uptake estimation methods. This should be completed in the methodological part
